# COMBINING LEARNED REPRESENTATIONS FOR COMBINATORIAL OPTIMIZATION

## ABSTRACT

We propose a new approach to combine Restricted Boltzmann Machines (RBMs) that can be used to solve combinatorial optimization problems. This allows synthesis of larger models from smaller RBMs that have been pretrained, thus effectively bypassing the problem of learning in large RBMs, and creating a system able to model a large, complex multi-modal space. We validate this approach by using learned representations to create "invertible boolean logic", where we can use Markov chain Monte Carlo (MCMC) approaches to find the solution to large scale boolean satisfiability problems and show viability towards other combinatorial optimization problems. Using this method, we are able to solve 64 bit addition based problems, as well as factorize 16 bit numbers. We find that these combined representations can provide a more accurate result for the same sample size as compared to a fully trained model.

## 1 INTRODUCTION

The Ising Problem has long been known to be in the class of NP-Hard problems, with no exact polynomial solution existing. Because of this, a large class of combinatorial optimization problems can be reformulated as Ising problems and solved by finding the ground state of that system (Barahona, 1982; Kirkpatrick et al., 1983; Lucas, 2014). The Boltzmann Machine (Ackley et al., 1987) was originally introduced as a constraint satisfaction network based on the Ising model problem, where the weights would encode some global constraints, and stochastic units were used to escape local minima. The original Boltzmann Machine found favor as a method to solve various combinatorial optimization problems (Korst & Aarts, 1989). However, learning was very slow with this model due to the difficulties with sampling and convergence, as well as the inability to exactly calculate the partition function. More recently, the Restricted Boltzmann Machine (RBM) has experienced a resurgence as a generative model that is able to fully approximate a probability distribution over binary variables due to its ease of computation and training via the contrastive divergence method (Hinton, 2002). The success of the RBM as a generative model has been limited due to the difficulties in running the Markov chain Monte Carlo (MCMC) algorithm to convergence (Tieleman, 2008; Tieleman & Hinton, 2009).

In this work, we propose a generative model composed of multiple learned modules that is able to solve a larger problem than the individually trained parts. This allows us to circumvent the problem of training large modules (which is equivalent to solving the optimization problem in the first place, as we are simply providing the correct answers to the module as training data), thus minimizing training time. As RBMs have the ability to fill in partial values and solutions, this approach is very flexible to the broad class of combinatorial optimization problems which can be composed of individual atomic operations or parts. Most notably, we show that our approach of using "invertible boolean logic" is a method of solving the boolean satisfiability problem, which can be mapped to a large class of combinatorial optimization problems directly (Cook, 1971; Karp, 1972). The ability of RBMs to fully model a probability distribution ensures model convergence and gives ideas about the shape of the underlying distribution. Many other generative models, such as Generative Adverserial Neural Networks (GANs), (Goodfellow et al., 2014) Generative Stochastic Networks (Alain et al., 2015) and others do not explicitly model the probability distribution in question, but rather train a generative machine to draw samples from the desired distribution. Although these can be useful for modeling high dimensional data, they do not provide the same guarantees that RBMs do. Because we use an RBM as our generative model we can perform a full Bayesian analysis, and condition on

any subset of the variables to solve a variety of problems. This leads to increased model flexibility, and the ability to generalize the learned models further.

## 2    RELATED WORK

People have shown that learned and trained features have the ability to outperform hand calculated features in a variety of tasks, from image classification to speech detection and many others. In addition, other network architectures, such as Recurrent Neural Networks (RNNs) have been shown to be Turing complete, and posses the ability to be used as a conventional Turing machine (Graves et al., 2014; Zaremba & Sutskever, 2014). However, these architectures have mostly been used as generalized computers, rather than to solve specific problems. Deterministic, feedforward neural networks have also been used to solve factorization problems, but their approach does not solve the factors for as high bit numbers as presented here, and is not flexible enough to be used in problems outside of prime factorization (Jansen & Nakayama, 2005). In addition, these models are not generative or reversible.

Many attempts have also been made to use quantum computers to solve such Ising model based problems. Large scale realizations of quantum computers, however, are still far from completion (Whitfield et al., 2012; Lucas, 2014; Roland & Cerf, 2002; Xu et al., 2012). Therefore, methods than can exploit a classical hardware are of critical importance.

In this regard, one recent work has proposed the use of "p-bits", to realize a form of Boltzmann Machine (Camsari et al., 2017). The work by (Traversa & Di Ventra, 2017) similarly tries to create an "invertible boolean logic", but does so in a deterministic manner, rather than the probabilistic one presented in this work.

Our approach falls within the broad category of transfer learning and compositional learning. There have been other works on using combinations of RBMs for object recognition and computer vision tasks (Ranzato et al., 2010), and on combinations of RBMs with weight sharing for collaborative filtering (Salakhutdinov et al., 2007). Nonetheless, the specific method presented here, as described in the following sections, to the best of our knowledge has not been used before.

## 3    APPROACH

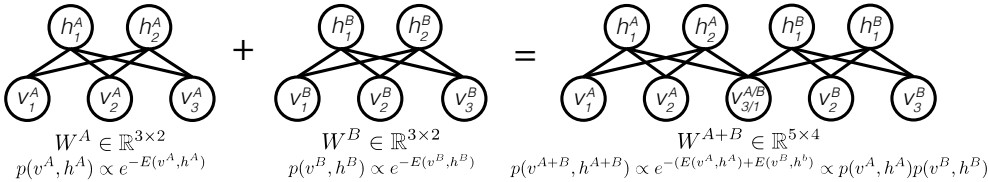

Figure 1: Merging procedure to combine RBMs. This combination scheme retains both the bipartite nature of the graph, as well as the product of experts nature. This allows the combined model to retain the sharp distributions of the original models.

An RBM is a binary stochastic neural network, which can be understood as a Markov random field of binary variables divided into a bipartite graph structure, where all of the visible units are conditionally independent of each other given the hidden unit states, and all hidden units are conditionally independent given the visible unit states. Here we denote $v$ as the visible state, $h$ as the hidden states, and $E(v, h)$ as the energy associated with those states. The probability assigned to a given state $p(v, h) = \frac{1}{Z} e^{-E(v,h)}$ where $Z = \sum_{v,h} e^{-E(v,h)}$ is the normalizing constant of the distribution.

The general approach to solving these combinatorial optimization problems is to recognize the atomic unit necessary to solve the problem, and use this as a base trained unit (this is discussed further later in the approach section). We combine these units by performing a merge operation of different RBMs. If we know two different units share a connection (such as the output of one logical unit being the input of another, or two cities being identical in a TSP), then these units would be merged to create a larger unit (see Figure 1 and explanation with weight matrices and biases below).

The probabilities and energies of the two RBMs are dictated by their weight matrices, $W_A \in \mathbb{R}^{n \times r}$ and $W_B \in \mathbb{R}^{m \times s}$, visible biases $b_A \in \mathbb{R}^n$ and $b_B \in \mathbb{R}^m$, and hidden biases $a_A \in \mathbb{R}^r$ and $a_B \in \mathbb{R}^s$. The energies and probabilities of these are as follows:

$$
\begin{aligned}
E_A(v,h) &= -v^T W_A h - a_A^T h - b_A^T v; \quad p_A(v,h) = \frac{1}{Z_A} e^{-E_A(v,h)} \\
E_B(v,h) &= -v^T W_B h - a_B^T h - b_B^T v; \quad p_B(v,h) = \frac{1}{Z_B} e^{-E_B(v,h)}
\end{aligned}
\tag{1}
$$

We can write the weight matrices as a series of row vectors corresponding to one visible units connections to a set of hidden units.

$$
W_A = \begin{bmatrix} -w_1^A- \\ \vdots \\ -w_n^A- \end{bmatrix}, \quad W_B = \begin{bmatrix} -w_1^B- \\ \vdots \\ -w_m^B- \end{bmatrix},
\tag{2}
$$

With this definition, the merge operation is shown below. If units $k$ of RBM $A$ is merged with unit $l$ of RBM $B$ the associated weight matrix $W_{A+B} \in \mathbb{R}^{(n+m-1) \times (r+s)}$, visible bias $v_{A+B} \in \mathbb{R}^{n+m-1}$ and hidden bias $h_{A+B} \in \mathbb{R}^{r+s}$ dictate the probabilities and energies for the merged RBM. Merging multiple units between these two RBMs corresponds to moving multiple row vectors from $W_B$ to $W_A$, which creates the associated decrease in dimensionality of $W_{A+B}$ and $b_{A+B}$ (where $W_{A+B} \in \mathbb{R}^{(n+m-d) \times (r+s)}$ and $v_{A+B} \in \mathbb{R}^{n+m-d}$ where $d$ is the number of merged units.

$$
W_{A+B} = \begin{bmatrix} W_A & \begin{matrix} 0 \\ -w_l^B- \\ 0 \end{matrix} \\ \hline 0 & W_{B \setminus l} \end{bmatrix} = \begin{bmatrix} \begin{matrix} -w_1^A- \\ \vdots \\ -w_k^A- \\ \vdots \\ -w_n^A- \end{matrix} & \begin{matrix} 0 \\ \\ -w_l^B- \\ \\ 0 \end{matrix} \\ \hline 0 & \begin{matrix} -w_1^B- \\ \vdots \\ -w_{l-1}^B- \\ -w_{l+1}^B- \\ \vdots \\ -w_m^B- \end{matrix} \end{bmatrix} \quad b_{A+B} = \begin{bmatrix} b_1^A \\ \vdots \\ b_k^A + b_l^B \\ \vdots \\ b_n^A \\ b_1^B \\ \vdots \\ b_{l-1}^B \\ b_{l+1}^B \\ \vdots \\ b_m^B \end{bmatrix} \quad a_{A+B} = \begin{bmatrix} a_1^A \\ \vdots \\ a_r^A \\ a_1^B \\ \vdots \\ a_s^B \end{bmatrix}
\tag{3}
$$

Below, we show how this relates to the original energies and probabilities. The vectors $v$ and $h$ corresponds to the visible vector put into the combined RBM, while $v_A$, $v_B$, $h_A$ and $h_B$ correspond to the equivalent state vectors that would be inputted into the single RBMs. Using these equations, we can see that the combined RBM energy factorizes into a sum of the original RBM energies and the probability is the product of the original probabilities.

$$v = \begin{bmatrix} v_1 \\ \vdots \\ v_{n+m-1} \end{bmatrix} \quad h = \begin{bmatrix} h_1 \\ \vdots \\ h_{r+s} \end{bmatrix} \quad v_A = \begin{bmatrix} v_1 \\ \vdots \\ v_l \\ \vdots \\ v_n \end{bmatrix}, v_B = \begin{bmatrix} v_{n+1} \\ \vdots \\ v_{n+l-1} \\ v_l \\ v_{n+l} \\ \vdots \\ v_{n+m-1} \end{bmatrix}, h_A = \begin{bmatrix} h_1 \\ \vdots \\ h_r \end{bmatrix} \quad h_B = \begin{bmatrix} h_{r+1} \\ \vdots \\ h_{r+s} \end{bmatrix}$$

$$(4)$$

$$E_{A+B}(v,h) = E_A(v_A, h_A) + E_B(v_B, h_B), \tag{5}$$

$$p_{A+B}(v,h) = \frac{1}{Z_{A+B}} e^{-E_{A+B}(v,h)} \propto p_A(v_A, h_A) p_B(v_B, h_B) \tag{6}$$

An alternative way of viewing the merged probability distribution is to multiply the distributions of the original RBMs, and marginalize over all states where the merged units are different.

Because of the probabilities approximately multiplying, we can also say that if each of the distributions differed from the "ideal" distribution (denoted here by $q$), then we can expect the error (as measured by the KL divergence) to increase approximately linearly with the number of connections. This shows that the error should increase with the number of connected units, rather than directly with the number of bits. This is especially important with combinatorial optimization problems where the deterministic algorithm run time increases exponentially with the number of bits. However, we note that as the number of connections between the units increases, the KL divergence increasingly diverges from linear.

$$D_{\mathrm{KL}}(q\|p) \approx D_{\mathrm{KL}}(q_A\|p_A) + D_{\mathrm{KL}}(q_B\|p_B);$$

$$p = p_A(v_A, h_A) p_B(v_B, h_B), \quad q = q_A(v_A, h_A) q_B(v_B, h_B)$$

Many combinatorial optimization problems can be broken down into associated sub-problems, and solved using a greedy approach (i.e. using the nearest neighbor approach in the Travelling Salesman Problem, or multiplying single digits in a larger multiplication, or evaluating one Boolean logic statement in a Boolean satisfiability problem). Using a greedy approach (such as used in the travelling salesman problem with nearest neighbors) can produce non-optimal results, and evaluating all permutations of possible solutions to find the most optimal can be computationally intractable in a large problem space. By combining these sub-problems using the method proposed here, we bypass the problems associated with those two approaches. We encode possible solutions as a probability indicating its local optimality, and combine these sub problems by merging the visible units of their RBMs as shown. The approach is that each sub-problem has overlapping units that share the same value, allowing them to be combined. This combination mechanism multiplies the probabilities such that the solution with global optimality is encoded as the mode of the distribution modeled by the larger RBM. As the combination method shown here still keeps the less optimal local solutions as a possible part of a global solution, we effectively bypass the problem of locally optimal solutions not solving the globally optimal problem. In addition, as the phase space of the problem space is $2^v$ where $v$ is the number of visible units, we can also encode a large problem space using minimal units and decrease the amount of computation. In this work we show a usage of this structure to solve a variant of the Boolean satisfiability problem by combining multiple trained logic gates, and use this to perform integer factorization and other arithmetic tasks. As the Boolean satisfiability problem can be mapped to solving a variety of NP-Hard problems (Cook, 1971; Karp, 1972), we believe variants of this approach can be used to solve other tasks. As an example, a usage for this approach to solve the Travelling Salesman Problem (TSP) might be to have an atomic unit that is the one step (or few) transitions between cities, and the global problem solved by combining these one (or few) step transition RBMs to create a full tour.

Solving a given optimization problem is equivalent to sampling from the model distribution of the RBM and finding the mode of that distribution. The RBM is partially clamped to values given in the problem statement, and samples are generated via MCMC as explained and examined below.

### 3.1 Convergence of MCMC

As sampling from the full distribution of an RBM is intractable due to the the partition function, a Markov chain Monte Carlo based technique is used to generate samples from the model distribution during inference both while training and while solving. Due to the bipartite graph nature of RBMs, Gibbs sampling is computationally efficient and can be done in 2 steps. Gibbs sampling converges to the model distribution in a geometric number of steps (Bremaud, 1999). Exact estimates of the convergence rate is intractable, as it involves calculating the SLEM (Second Largest Eigenvalue Modulus) of the gibbs transition matrix. However, a theoretical bound can be analyzed using Dobrushin's Ergodic Coefficient (Dobrushin, 1956) as shown below. A similar derivation is also shown in Bremaud (1999).

$$|\mu P^n - \pi| \leq \frac{1}{2}|\mu - \pi|(1 - e^{-2\Delta})^n \tag{7}$$

$$\Delta = \sup_{x,y \in \{0,1\}^N} \{|E(x_v, x_h) - E(y_v, y_h))|\}$$

A proof of this is shown in the Appendix, Section 6.1. In this equation, $\mu$ is the starting distribution (or starting point) $P$ is the transition probability matrix (represented by applying the gibbs transition operator on all units), $\pi$ is the stationary distribution and $n$ is the number of transition steps taken. The convergence parameter $\Delta$ represents the maximum energy difference between two states in the RBM, where we denote the to states $x$ and $y$, with visible and hidden states $(x_v, x_h)$ and $(y_v, y_h)$ respectively.

From our derivation above, we can estimate the effects of merging on the upper bound of convergence rate. If we combine two RBMs $A$ and $B$, with parameters labeled as such the highest energy states correspond to "correct" answers ($E_A^{max}$, $E_B^{max}$) and the lowest energy states correspond to "incorrect" answers ($E_A^{min}$, $E_B^{min}$). Denote the maximum and minimum energy of the combined RBM as $E_{comb}^{max}$ and $E_{comb}^{min}$, and $\Delta_{comb}$ as the convergence coefficient for the combined RBM.

$$E_{comb}^{max} \leq E_A^{max} + E_B^{max}; \quad E_{comb}^{min} \geq E_A^{min} + E_B^{min}$$

$$\Delta_{comb} \leq \Delta_A + \Delta_B$$

$$|\mu P^n - \pi| \leq \frac{1}{2}|\mu - \pi|(1 - e^{-2\Delta_{comb}})^n \leq \frac{1}{2}|\mu - \pi|(1 - e^{-2(\Delta_A + \Delta_B)})^n$$

This implies a considerable decrease in convergence rate over the original RBMs, amounting to an exponential increase in the upper bound of convergence constant as more RBMs are combined together. We note that this is meant to be a theoretical upper bound on convergence, and thus the bound is never tight. Empirical results show that in many cases, this bound tends to be very loose, even while merging. For this reason, an empirical study is necessary to understand the effects of merging RBMs. We explain some of the empirical effects and their causes below.

The autocorrelation coefficient ($AR(k)$) is a good proxy for empirically measuring rate of convergence, as it measures how correlated samples from subsequent steps are. When $AR(k) \approx 0$ samples separated by $k$ steps are expected to be independent. There are many things that can effect the convergence rate of the Markov chain, such as the magnitude of the weight matrices, the modality of the phase space, and the sparsity of the weight matrix.

### 3.1.1 WEIGHT MATRIX MAGNITUDE

Large magnitude weights and biases create higher and lower energy states, and thus a higher $\Delta$. This is a well known phenomenon, and effects RBMs during both training and inference. The magnitude of the weights can be controlled by using finite weight decay during training (Hinton, 2002; Tieleman, 2008).

### 3.1.2 STATE SPACE & MODEL MODALITY

Many combinatorial optimization problems are highly multimodal, which can lead to problems of getting stuck in local minima and spurious modes during MCMC. This problem can manifest itself in a large combined structure, where there are many low energy states caused by the low energy states of the individual RBMs. Once the MCMC algorithm finds a low energy mode it tends to stay around it. To leave the mode, the Markov Chain must do a traversal from one of these modes to another, which goes as $p^n$ where $p$ is the probability of transition between states and $n$ is the number of transitions needed to go from one mode to another. As the distribution becomes increasingly multimodal, pseudo-convergence of the MCMC becomes a problem.

### 3.1.3 SPARSITY

A very sparse weight matrix (as is pictured in Figure 2) can cause a relatively sparse 1 step transition probability matrix while running MCMC (we note that a precisely 0 transition would only occur with infinite energies, we mean that the probability of transitioning between two states becomes small). This means that for visible units on the either end of the model to mix, they would have to cross a large number of intermediate states. This can be seen in figure 2 where in the 16 bit adder composed of multiple one 1 bit adders, there is no hidden unit directly connecting the first full adder to the last full adder. Thus, for information about the change of state in the first full adder must pass through all of the individual adders before it can manifest itself as a change in the last full adder. This means additional gibbs sampling steps must be taken, causing a slower convergence rate for the RBM during inference. This effect is partially mitigated by the fact that very sparse matrices can be more computationally efficient for matrix multiplications, causing the actual computation time to be less.

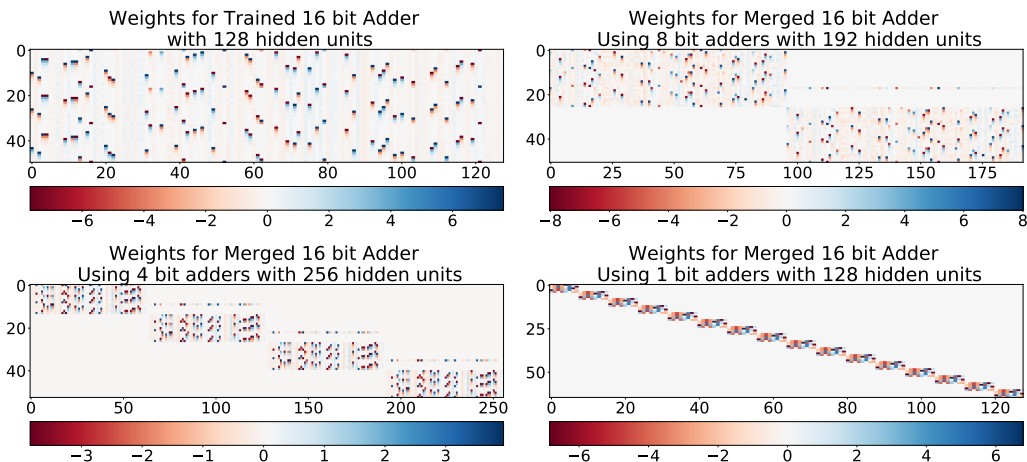

Figure 2: Weights matrices for a 16 bit adder circuit created by directly training the device (top left), combination of 8 bit units (top right), combinations of 4 bit units (bottom left) and combination of 1 bit units (bottom right). The $x$ axis enumerates hidden units, and $y$ axis visible units. The sparsity can cause matrix multiplication to be more efficient, but can also lead to poor mixing between states.

## 3.2 TRAINING VS. SETTING MODEL PARAMETERS

If we are aware of the exact distribution we are trying to model, we can also directly set the weight matrix and biases for the RBM instead of training using samples from this exact distribution. As the RBM is a universal function approximator, we are guaranteed to have the correct distribution for an arbitrary sized model. Using the method outlined in the proof of universal function approximation in (Le Roux & Bengio, 2008) we can create an "ideal" RBM for this problem, and merge copies of these ideal RBMs together. We note that using these directly calculated RBMs can allow for further guarantees for convergence rates, but suffers in other ways (Bremaud, 1999).

There are a few problems with this method, most notably that directly calculated RBMs are not compact function approximators. To exactly model these distributions, 1 hidden unit is needed to approximate every data vector meaning the number of hidden units would scale as $2^n$ for multiplier and adder units. This type of scaling is not expected for most highly structured problems, as many combinatorial optimization tasks are. The directly calculated RBM is more susceptible to being trapped in local minima than a trained RBM is due to poor mixing between modes. This is because they have a uniformly low probability in states that are not exactly part of the data distribution as described above. This is also seen in Figure 4 where the directly calculated unit has a slower convergence rate, owing to the slower decay to an autocorrelation of 0.

## 4 EXPERIMENTS

To experimentally demonstrate the viability of this method of combination, we have used it to create "invertible boolean logic", which is a form of solving the boolean satisfiability problem. With this, we are able to create logical units of varying sizes, and combine them to create adders and multipliers. We could also combine these logical units to solve other types of boolean satisfiability problems, but we do explain that further here. Using the property of RBMs that they can partially fill in visible units conditioned on the values of other units, we can use these adder RBMs to solve addition, subtraction, reverse sum carry, and combine these adders with multiplier RBMs to solve multiplication, division, and factorization tasks. Precisely, we train the RBMs on the joint distribution of $n$-bit addition (or multiplication), then clamp some of the values to data and perform gibbs sampling on the others conditioned on the partial data. We are able to use a circuit that is traditionally used to multiply numbers to also divide, factorize, and solve any problem involving partial multiplication and division, as we are just sampling from the joint distribution of variables over multiplication and conditioning over variables we are interested in.

Performance on these various tasks was characterized by using Gibbs sampling, and taking samples from the distribution conditioned on partial inputs. After a number of samples, we check the mode of the distribution against the expected minimal energy state. As described in Section 3 the sampled distribution converges in distribution to the model distribution in a geometric number of steps, at which point the mode is guaranteed to be correctly identified.

This merging method is validated by the equilibrium distributions of a merged adder vs. a trained adder in Figure 3. We compare the performance of an RBM created by merging individually trained logic units (AND, XOR, OR, etc.) to the performance of a full adder. When we sample from the full joint distribution, we are solving the boolean satisfiability problem on the network, by finding all possible solutions to the given boolean statement. The equilibrium distributions are very similar, but the two distributions mix very differently. The trained full adder mixes rapidly between modes while the full adder composed of merged logical units is more likely to get stuck in a one mode. This condition can be mitigated by starting multiple chains from various starting locations in the state space and combining their distributions using a type of mulistart heuristic (Brooks et al., 2011). However, it is better practice to design the weight matrix to have better mixing properties and to use more advanced sampling techniques other than Gibbs Sampling.

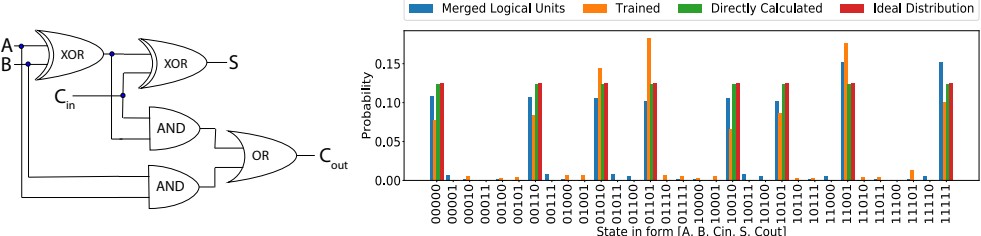

Figure 3: Probability distributions calculated for a trained full adder, as well as a full adder composed of merged logical units as shown on the left using the method in Section 3. The comparison shows that merging units can approach similar performance to a fully trained unit, with similar error levels even with a reasonable number of merged units. These probabilities were directly calculated and normalized by the exact partition function (no gibbs sampling was done). We also show directly calculated (as described in Le Roux & Bengio (2008)) units have the distribution closest to the ideal distribution. We note that this is exactly solving the boolean satisfiability problem over the boolean circuit shown on the left, and thus inherently solves the boolean satisfiability problem.

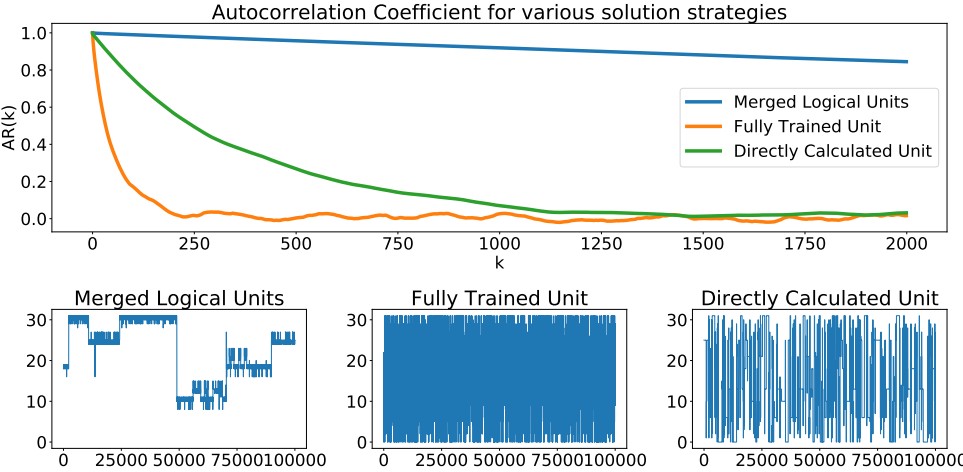

Figure 4: Autocorrelation coefficients (above) and time series (below) of MCMC algorithm comparison. All three units are attempting to model the behavior of a full adder circuit using either Merged Logical Units, a fully trained unit, or a directly calculated unit. The probability distribution modeled by all of these units is seen in Figure 3. We can see that the convergence rate for a fully trained unit is much faster than that of both a directly calculated unit and of merged logical units as the autocorrelation coefficient quickly decays to 0 for this case. From the time series analysis below we can also see that the merged logical units take a longer time to visit the entirety of the state space, as is expected by the autocorrelation coefficients.

## 4.1 ADDERS

Multi bit adders were created using the combination scheme outlined in Figure 5. From the data based on trained and merged 64 bit adders, we are unable to improve performance by training larger individual units. One of the biggest issues is the exponential growth in the size of the training space (and time) needed to effectively train larger units. As the training set grows as $2^n$, training on the entirety of an addition dataset becomes intractable at 16 bit adders, where a training set that would encompass the entire space would contain $2^{33} \approx 8$ billion training examples. Even though we expect these models to generalize effectively, training a good model becomes increasingly difficult as the dataset grows. This can be seen in greater degree in the 32 bit adder unit, whose performance is significantly worse than merged units even after training the model for over a week. We attempted

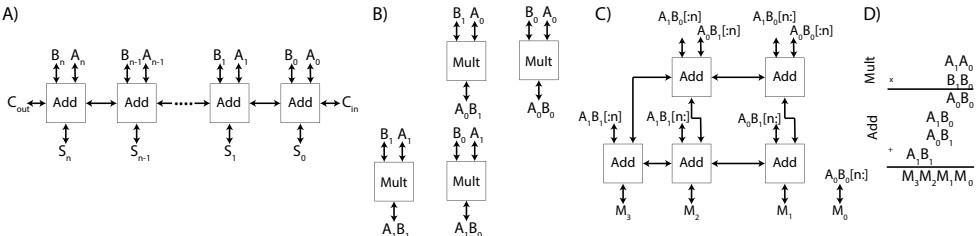

Figure 5: Diagram of the creation of larger logical units by combining smaller individual units. We can create a $n$-bit adder can be created by cascading $n$ copies of a 1 bit adder (or $n/2$ copies of a 2 bit adder etc.) as in A). To create a multiplier, we divide n bit multiplication as described in D) into $n/2$ bit multiplication by dividing the problem into multiplication as in B) along with addition of partial sums, as in C). This can also be extended to create $n$-bit multiplication using smaller units.

to train a 64 bit adder unit but were unsuccessful in reducing the error by any reasonable amount, so these results were not included.

We note the relative success of merging addition models in comparison to multiplication models. This can be explained by the fact that addition models are less prone to getting stuck in local minima, as each of the adders is clamped to a piece of the data in all modes of operation (addition, subtraction and reverse sum carry). The 64 bit adder unit can explore a state space of $2^{64} \approx 2 \times 10^{19}$ states, and find the exact answer within only $\approx 10^3$ samples, showing a remarkably fast convergence rate. We also note that the adders converge significantly faster than the upper bound of the convergence rate given in Section 3.1.

## 4.2 MULTIPLIERS

Multi bit multipliers were created by the combination scheme outlined in Figure 5, similar to Adders. Multipliers require heterogeneous integration of different components (adders and multipliers), which means that the magnitude of multiple different weight matrices need to be matched to get good mixing and to ensure there are minimal spurious modes.

The design of bitwise multipliers faces more challenging issues than that of adders. The state space of multipliers is much sparser, tends to have more spurious modes, is larger for the same number of bits. For an adder, the state space is $2^{3n+2}$ states ($2 \times n$ inputs, $n$ output, 1 $C_{in}$, 1 $C_{out}$) with $2^{2n+1}$ of those states representing valid answers to the addition problem. However, the multiplier has a state space that is $2^{4n}$ ($2 \times n$ inputs, $2 \times n$ outputs) and $2^{2n}$ of them are valid answers. This means that an 8 bit multiplier has a state space that is 128 times sparser than an 8 bit adder, and that the sparsity of an adder scales as $\approx 2^n$ while the sparsity of a multiplier scales as $\approx 2^{2n}$. This increased sparsity makes mixing between modes more difficult, as there are larger areas of low probability space in between modes.

Multipliers have a more complex state space that leads them to get stuck in spurious modes. This is due to the fact that not all units in the multiplier are clamped to pieces of the data distribution, and intermediate units have many equivalent local minima. Because of this, individual units may be in local minima where their individual constraints are satisfied (i.e. an adder satisfies the condition that its inputs add to its outputs, or a multiplier satisfies the constraints that the output is equal to the two inputs are multiplied together), but the global constraint that the output is equal to the inputs multiplied together is unsatisfied. This can be because one of the units is in an incorrect state, or the local minima of the other units corresponds to an unsatisfiable constraint on one of the units (i.e. an adder that is forced into a state where the numbers cannot add up to one another, or a multiplier that is forced to factorize a number whose factors are outside the input bit range).

Even with the complexities of multipliers, we have shown that an 16 bit multiplier created from trained and merged is able to outperform a multiplier created just by training, as can be seen in Figure 6. Currently, the number of samples needed to compute the factors is not competitive with a direct search. However, we believe that this situation will be significantly improved by engineering of the convergence rate of the MCMC algorithm, and using more advanced sampling techniques (as mentioned below in Section 5).

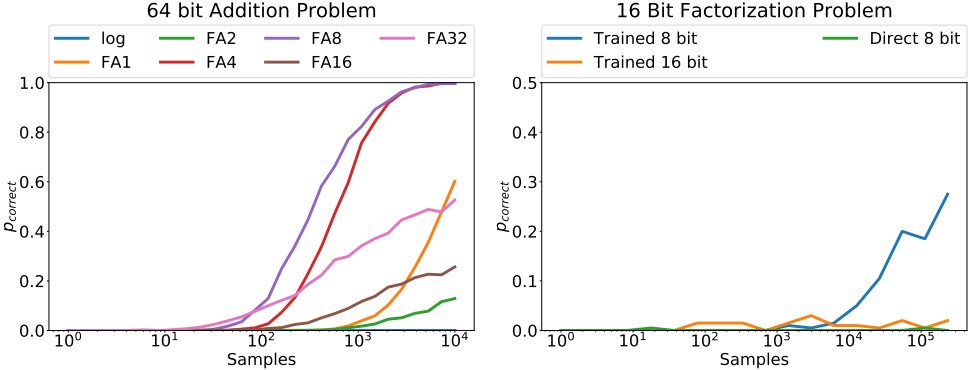

Figure 6: In both the factorization and addition problems, as we approach higher bit problems it becomes increasingly difficult to fully train a module. At 16 bits for factorization (8 bit factors, 16 bit number) and 64 bits for addition (64 bit inputs), it becomes clear that merged logical units can outperform fully trained units. There is a clear trade off between the ability to train the model and number of samples needed to perform accurate inference. The legends indicate the number of bits the base trained model (before combination) is using, i.e. "FA8" corresponds to combined 8 bit adders, "FA4" is 4 bit adders, "log" refers to merged logical units. Further results are presented in the Appendix, Section 6.3

## 5 CONCLUSION & FUTURE WORK

In this work we have shown the viability of merging RBMs of individually trained distributions and using this merged RBM to solve combinatorial optimization tasks. Although mixing in these merged RBMs is a problem, these merged representations can outperform fully trained structures on tasks where training is difficult. We have used boolean satisfiability and factorization as an example problem to validate this technique. However, the approach is generally applicable to a variety of other combinatorial optimization tasks.

One of the biggest challenges in making this approach viable as a stochastic optimization algorithm is to improve the convergence rate of the merged RBMs. In addition to the weight decay approaches already implemented in this work, significant improvement in this regard is expected from using advanced sampling techniques such as tempered transitions (Desjardins et al., 2010), fast weight decay (Tieleman & Hinton, 2009), adaptive MCMC (Salakhutdinov, 2010), annealed importance sampling (Neal, 2001), parallel tempering (Cho et al., 2010), or using a different transition operation for the RBM (Brgge et al., 2013). It will also be necessary to develop mathematical models that can describe the differences in the convergence rate of the merged model as compared to that of the constituent units.

RBMs have been used for classical combinatorial optimization problems such as the travelling salesman (Shim et al., 2012; Aarts & Korst, 1989). The method presented here is generally applicable to all such combinatorial optimization tasks. We have outlined how such an approach could be done in Section 3. As a specific example, a natural application for the adders, that we used as an example case, could be to solve 3SUM and subset sum problems . There have already been initial results on these problems using cascaded, fully connected Boltzmann Machines (Hassan

et al., 2018).

ACKNOWLEDGMENTS

This work was funded through the ASCENT center, one of the six centers within the DARPA/SRC JUMP initiative.

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

## 6 APPENDIX

### 6.1 PROOF OF EQUATION 7

This proof follows a similar structure to Bremaud (1999), but has further simplifications and additions due to the conditional independence structure of the RBM.

Gibbs Transitions over the RBM can be be factored into two stages, a transition probability distribution over visible units, and a transition probability distribution over hidden units. The respective transition matrices are multiplied to yield the full state transition matrix. We will call $\mathbf{P}$ the full transition matrix with elements $p_{xy}$, and $\mathbf{P}_v$, $\mathbf{P}_h$ the visible and hidden transitions with elements $p_{xy}^v$ and $p_{xy}^h$ respectively.

$$\mathbf{P} = \mathbf{P}_v \mathbf{P}_h \tag{8}$$

$$p_{xy}^v = \frac{e^{-E(y_v, x_h)}}{\sum_v e^{-E(v, x_h)}}, \quad p_{xy}^h = \frac{e^{-E(x_v, y_h)}}{\sum_h e^{-E(x_v, h)}} \tag{9}$$

The states $x$ and $y$ are parameterized as $x = (x_v, x_h)$ with visible ($x_v$) and hidden ($x_h$) portions. The individual transition probabilities $p_{xy}^v$ represent the probability of transitioning from state $x$ to state $y$, when only changing the visible states of $x$. Thus, the entry $p_{xy}^v$ in $\mathbf{P}_v$ is nonzero iff $x_h = y_h$.

Starting with the convergence inequality from Dobrushin (1956).

$$|\mu P^n - \pi| \leq \frac{1}{2}|\mu - \pi|(\delta(\mathbf{P})^n$$

$$\delta(\mathbf{P}) = 1 - \inf_{x,y \in \{0,1\}^N} \sum_{k \in \{0,1\}^N} \min(p_{xk}, p_{yk})$$

From here, we can take further bounds on the ergodic coefficient:

$$\delta(\mathbf{P}) \leq 1 - 2^N \inf_{x,y} p_{xy}$$

Defining $m_v(h) = \inf_{v \in \{0,1\}^{N_v}} (E(v, h))$ and $m_h(v) = \inf_{h \in \{0,1\}^{N_h}} (E(v, h))$, these represent the states with lowest transition probability from a given input state. The variables $N_v$ and $N_h$ represents the number of visible and hidden units respectively.

$$p_{xy}^v = \frac{e^{-(E(y_v, x_h) - m_v(x))}}{\sum_v e^{-(E(v, x_h) - m_v(x))}} \geq \frac{e^{-\delta_v}}{2^{N_v}}$$

$$\delta_v = \sup_{h \in \{0,1\}^{N_h}} (|E(v', h) - E(v, h)|; v, v' \in \{0,1\}^{N_v})$$

$$p_{xy}^h = \frac{e^{-(E(x_v, y_h) - m_h(x))}}{\sum_h e^{-(E(x_v, h) - m_h(x))}} \geq \frac{e^{-\delta_h}}{2^{N_h}}$$

$$\delta_h = \sup_{v \in \{0,1\}^{N_v}} (|E(v, h') - E(v, h)|; h, h' \in \{0,1\}^{N_h})$$

In words, $\delta_v$ is the maximal energy difference between two states that have the same hidden states, and $\delta_h$ is the maximal energy difference between two states that have the same visible states. Using this definition, we can define

$$\inf_{x,y} p_{xy} \geq \frac{e^{-(\delta_v + \delta_h)}}{2^N} \geq \frac{e^{-2\Delta}}{2^N}$$

$$\Delta = \sup_{x,y \in \{0,1\}^N} \{|E(x_v, x_h) - E(y_v, y_h))|\}$$

And finally

$$\delta(\mathbf{P}) \leq 1 - 2^N \inf_{x,y} p_{xy} \leq 1 - e^{-2\Delta}$$

$$|\mu P^n - \pi| \leq \frac{1}{2}|\mu - \pi|(\delta(\mathbf{P}))^n \leq \frac{1}{2}|\mu - \pi|(1 - e^{-2\Delta})^n$$

## 6.2 TRAINING

In this paper we trained the RBMs by contrastive divergence as described by Hinton (2002). Each of the models in this paper were validated by checking their performance on the problem they were trying to solve (i.e addition and subtraction for an adder, multiplication and factorization for a multiplier). This method was also used to assess model complexity (i.e number of hidden units) and evaluate learning parameters (learning rate, batch size, etc.). The final results for RBM sizes are shown below.

Training was conducted on a computer with 2 Intel Xeon E5-2620 processors, and 2 Nvidia Titan V GPUs with 128Gb RAM. Each RBM was trained for 10 epochs, where an epoch was 4 copies of the full state space, with a learning rate of 1. After 10 epochs, the CDk of learning was increased to combat the worse mixing as the weight matrix increases, with an initial CDk of 2. This process was repeated until the test error stopped decreasing. In the case that the state space was too large to train on all of it (such as the 16 bit and 32 bit adders), a random sample of the training set was used, and a new random sample was reinitialized every epoch of training.

Table 1: Trained RBM Parameters

| RBM | Hidden Units | Training Time (minutes) |
|---|---|---|
| 1 bit Adder | 6 | 1 |
| 2 bit Adder | 28 | 13.5 |
| 4 bit Adder | 64 | 133 |
| 8 bit Adder | 96 | 201 |
| 16 bit Adder | 128 | 321 |
| 32 bit Adder | 192 | 13000 (approx) |
| 1 bit Multiplier | 4 | 1 |
| 2 bit Multiplier | 12 | 46 |
| 4 bit Multiplier | 64 | 655 |
| 8 bit Multiplier | 96 | 2794 |

The training time tends to increase with the number of bits in the adder or multiplier due to both the size of the data set (which increases exponentially with the number of bits) and the number of hidden and visible units (which both increase approximately linearly). The slower increase of training time for adders after the 4 bit adder is due to the usage of GPUs to increase the parallelism.

At the 16 bit adder level, the size of the data set was so large that the entire data set could not be used for training ($\approx 8$ billion data points) and a randomized sample of the set had to be taken. As generalization is not perfect, we can attribute the decrease in their performance (as described in Figures 6 8 7) to this fact. For the 32 bit adder, this problem was exacerbated, and the 32 bit adder was outperformed by most units even after training for a full week.

For multipliers, the 8 bit multiplier has a tractable amount of data, but a good joint density model could not be formed even after a large training time. We believe this is due to an inherent difficult in the multiplication problem that is not present in the addition problem. As there is not as distinct of a correlation between higher level bits in the 8 bit multiplication problem as there is in the addition problem, first level correlations (as an RBM with 1 layer of hidden units would find) are more difficult to find. We believe that using deep boltzmann machines might help fix the problem of training in large multipliers.

## 6.3 FURTHER DATA

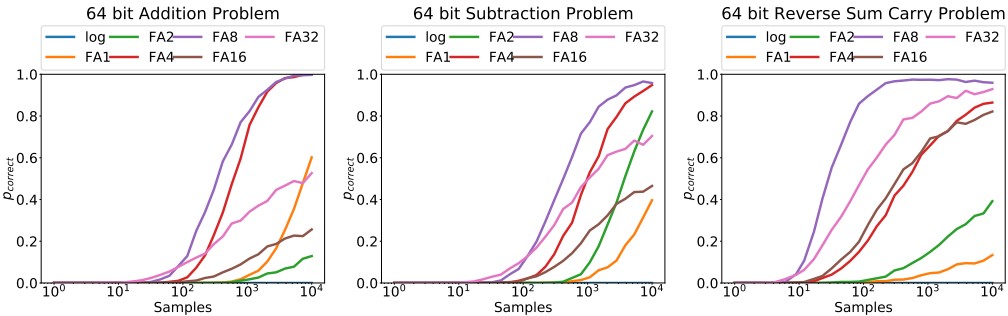

Figure 7: A comparison of probability of being correct vs. number of samples taken for a 64 bit adder unit composed of smaller merged units. It is clear from these figures that smaller bit units that can be trained better outperform larger, harder to train units.

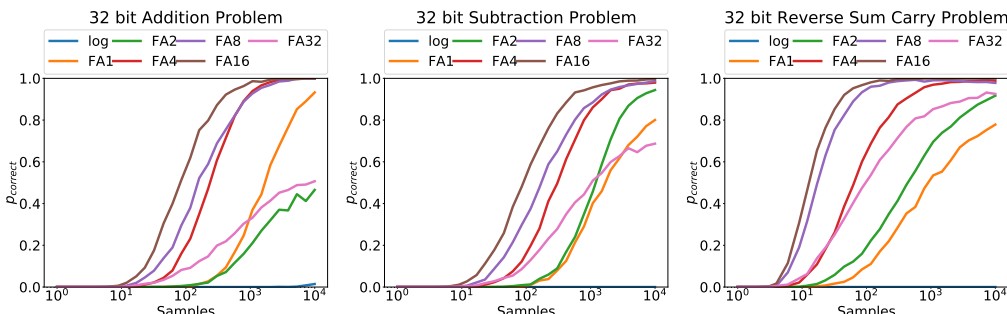

Figure 8: A comparison of probability of being correct vs. number of samples taken for a 32 bit adder unit composed of smaller merged units compared to a fully trained model. The joint density model for the 32 bit adder is an intractable training problem due to the size of the data set ($2^{66}$ datapoints).

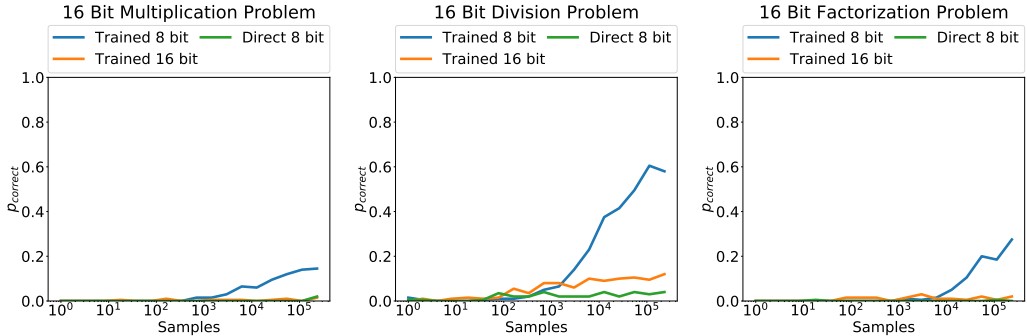

Figure 9: A comparison of probability of being correct vs. number of samples for an 16 bit multiplier unit. Here we compare using a fully trained unit, vs. composing it of 8 bit trained multiplier and adder units, vs. composing it with 8 bit directly calculated unit. The trained unit outperforms both of these units, even after extensive training of the 16 bit multiplier unit (see above 6.2)

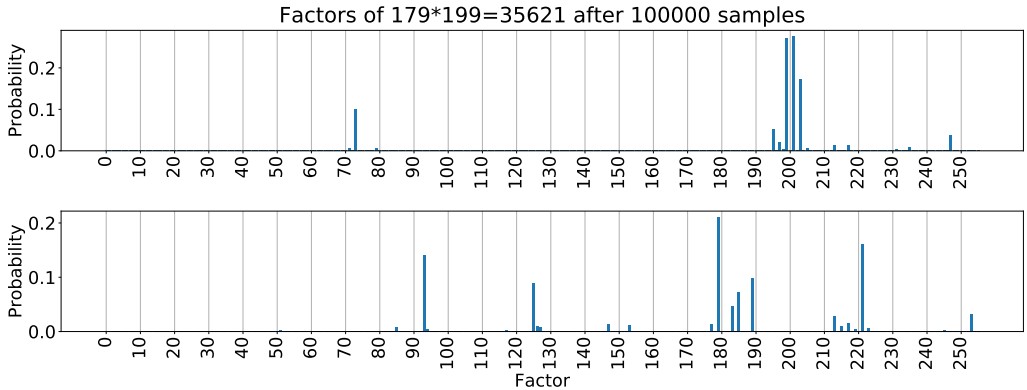

Figure 10: An example output from a factorization problem for the merged 8 bit multiplier formed from 4 bit units. This is an example of the probability distribution generated by MCMC after 100000 samples. In this example, we are trying to find the coprimes that multiply to 35621. The factors it should find are 179 and 199, which it correctly identifies as the most likely factors.

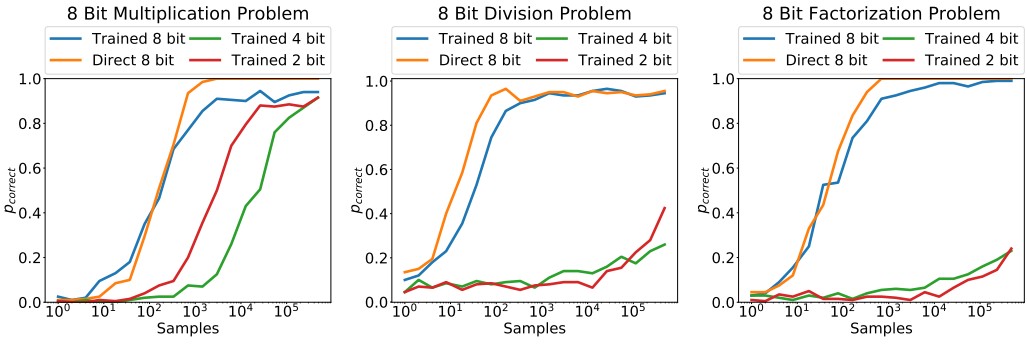

Figure 11: A comparison of probability of being correct vs. number of samples for an 8 bit multiplier unit. Training of an 8 bit multiplier is a tractable problem, and we can see that both the trained and the directly calculated models outperform the merged models. This can be understood as a consequence of worse mixing present in the merged models. (see above 6.2)

