# OpenReview forum: "Combining Learned Representations for Combinatorial Optimization"
_ICLR.cc/2019/Conference_

### Official Review · AnonReviewer2 · 2018-10-28
**The novelty and supporting theory is not significant**

**Rating:** 5
**Confidence:** 3

**Review:**

The paper introduces a new approach to combine small RBMs that are pretrained in order to obtain a large RBM with good performance. This will bypass the need of training large RBMs and suggests to break them into smaller ones. The paper then provides experimental evidence by applying the method on "invertible boolean logic". MCMC is used to find the the solution to large RBM and compare it against the combined solutions of smaller RBMs.


The paper motivates the problem well, however, it is not well-written and at times it is hard to follow. The details of the approach is not entirely clear and no theoritcal results are provided to support the approach. For instance, in the introduced approach, only an example of combination is provided in Figure 1. It is not clear how smaller RBMs (and their associated parameters) are combined to obtain the larger RBM model. From the experimental perspective, the experimental evidence on "invertible boolean logic" does not seem to be very convincing for validating the approach. Additionally, the details of the settings of the experiments are not fully discussed. For example, what are the atomic/smaller problems and associated RBMs? what is the larger problem and how is the corresponding RBM obtained? Overall, the paper seems to be a report consisting of a few interesting observations rather than introducing a solid and novel contribution with theoretical guarantees.

Remark:
The term "Combinatorial optimization", which is used in the title and throughout the body of paper, sounds a bit confusing to the reviwer. This term is typically used in other contexts.

Typos:
** Page 2 -- Paragraph 2: "Therefore, methods than can exploit..."
** Page 3 -- 2nd line of math: Super-scripts are missing for some entries of the matrices W^A and W^{A+B}
** Page 5 -- Last paragraph: "...merged logical units is more likly to get get stuck in a ..."
** Page 5 -- Last paragraph: "...and combining their distributions using the mulistart heuristic..."

---

> ### Author Response · Authors · 2018-11-25
> **Response to AnonReviewer2**
>
> Thank you for your comments, we will be responding with specific comments to AnonReviewer3 here, and more general comments to the reviewer above.
>
> R2: For instance, in the introduced approach, only an example of combination is provided in Figure 1. It is not clear how smaller RBMs (and their associated parameters) are combined to obtain the larger RBM model.
>
> As far as explaining the method of combination, and the associated mathematical properties, we have tried to do this in greater detail in section 3 (Approach). We used Figure 1 and the combination matrices to show what exactly is happening when we combine the models, and how the models mathematically combine. In our revision we have made an effort to outline this in greater detail.
>
> R2: Overall, the paper seems to be a report consisting of a few interesting observations rather than introducing a solid and novel contribution with theoretical guarantees.
>
> In regards to the lack of theoretical guarantees, we have shown that the equilibrium distribution is what we expect it to be, and mathematically have shown that the final distribution of interest has the mode we expect it to. It has been shown in many texts that Gibbs Sampling converges to this equilibrium distribution at a geometric rate in Markov Random Fields.  Finding the exact convergence rate involves calculation of the eigenstructure of the markov chain transition matrix, which is in general computationally intractable for RBMs of moderate size [1]. Given this, we have added an extra theorem to show how the upper bounds on convergence rate changes as we merge RBMs, this can be seen in Section 3.1 on “Convergence Rate and MCMC”. We show that the rate of convergence of the RBM is geometric in the number of sampling steps, and that the combined RBM will have a convergence rate bounded by the sum of the convergence rates of the individual RBMs.
>
>  If we want to have further theoretical guarantees, we have the ability to exactly set model parameters, as mentioned in section 3.2 to get the exact distribution of interest, and to combine those RBMs with directly calculated parameters. As mentioned in that section, this is not a data efficient, or computationally efficient method which is why we chose to not pursue it.
>
> [1] Pierre. Bremaud. Markov Chains: Gibbs Fields, Monte Carlo Simulation, and Queues, volume 1.Springer New York, 1999. ISBN 9781441931313.

---

### Official Review · AnonReviewer3 · 2018-11-02
**Need experiments on more challenging tasks**

**Rating:** 4
**Confidence:** 5

**Review:**

The paper proposes learning Restricted Boltzmann Machines for solving small computational tasks (e.g., 1-bit addition) and composing those RBMs to form a more complex computational module (e.g., 16-bit addition). The claim is that such an approach can be more data efficient than learning a single network to directly learn the more complex module. Results are shown for addition and factoring tasks.

- The paper is somewhat easy to follow and the figures are helpful. But the overall organization and flow of ideas can be improved significantly.
- The term "combinatorial optimization" is used in a confusing way -- addition would not usually be called a combinatorial optimization problem.
- It would be good to understand what benefit does the stochasticity of RBMs provide. How do deterministic neural networks perform on the addition and factoring tasks? The choice of RBMs is not motivated well and without any comparisons to alternatives, it comes across as arbitrary.
- That learning simple functions and composing them to compute more complex functions would be more data efficient than directly learning the complex functions does not seem very surprising.  After all, the former approach gets a lot more knowledge about the target function built into it. It's good that the paper empirically confirms the intuition, but doesn't feel like a significant contribution on its own.
- The paper would be stronger if it includes more complex tasks, e.g., TSP, and show that the same ideas can be applied to improve the learning a solver for such tasks. The current tasks and problem sizes are not very convincing, and the accuracy results are not very compelling.

---

> ### Author Response · Authors · 2018-11-25
> **Response to Reviewer**
>
> Thank you for your comments, we will be responding with specific comments to AnonReviewer3 here, and more general comments to the reviewer above.
>
> R3: “It would be good to understand what benefit does the stochasticity of RBMs provide.”
>
> The stochasticity of the RBM provides a number of benefits over more deterministic methods. Firstly, the stochasticity allows for full sampling from the RBMs distribution, and has the ability to identify all possible modes in a multimodal distribution if sampled for long enough.  As there are many possible solutions to a Boolean logic query (and integers can have many different factors), we note that the statistics that this method provides can give us a variety of answers to the queries, allowing the user to evaluate each individual solution based on its individual merit.
>
> R3: How do deterministic neural networks perform on the addition and factoring tasks? The choice of RBMs is not motivated well and without any comparisons to alternatives, it comes across as arbitrary.
>
> To the best of our knowledge, deterministic neural networks have not been well studied for the integer factorization problem. In [4] deterministic neural networks are used, but are able to factor smaller integers, on a more restricted problem, and are fully trained on the subset of all integers. We have cited this work in our related works section, and mentioned its impact.
>
> R3: That learning simple functions and composing them to compute more complex functions would be more data efficient than directly learning the complex functions does not seem very surprising.
>
> We agree that this method of composing simple functions to compute more complex ones is intuitive, and may not be very surprising, but we think that this helps data and model efficiency in a different manner than presented in previous papers.
>
> As far as scaling up the tasks and problem sizes, we are showing a method of combination here, and are scaling up the problem sizes continuously. We believe this combination method could be used for other things, and have presented it here as a proof of concept rather than a definitive survey with all possible uses.

---

### Official Review · AnonReviewer4 · 2018-11-09

**Rating:** 4
**Confidence:** 3

**Review:**

The paper proposes to combine several smaller, pretrained RBMs into a larger model as a way to solve combinatorial optimization problems. Results are presented on RBMs trained to implement binary addition, multiplication, and factorization, where the proposed approach is compared with the baseline of training a full model from scratch.

I found the paper confusing at times. It is well-written from a syntactical and grammatical point of view, but some key concepts are stated without being explained, which gives the impression that the authors have a clear understanding of the material presented in the paper but communicate only part of the full picture to the reader.

For instance, there’s a brief exposition of the connection between Boltzmann machines and combinatorial optimization problems: the latter is mapped onto the former by expressing constraints as a fixed set of Boltzmann machine weights and biases, and low-energy states (i.e. more optimal solutions) are found by sampling from the model, which involves no training. What’s less clear to me is what kinds of combinatorial optimization problems can be mapped onto the RBM *training* problem. The paper states that the problem of training "large modules" is "equivalent to solving the optimization problem", but does not explain how.

Similarly, the paper mentions that the "general approach to solving these combinatorial optimization problems is to recognize the atomic unit necessary to solve the problem", but at that point the reader has no concrete example of what combinatorial optimization problem would be mapped onto training and inference in RBMS.

A concrete example is provided in the Experiments section: the authors propose to implement invertible (reversible?) boolean logic circuits by combining smaller pre-trained RBMs which implement certain logical operations into larger circuits. I have two issues with the chosen example: 1) the connection with combinatorial optimization is not clear to me, and 2) it’s not very well explained. As far as I understand, these reversible boolean logic operations are expressed as sampling a subset of the RBM’s inputs conditioned on another subset of its inputs. An example is presented in Figure 3 but is not expanded upon in the main text. I’d like the authors to validate my understanding:

An RBM is trained to implement a complete binary adder circuit by having it model the joint distribution of the adder’s inputs and outputs [A, B, Cin, S, Cout] (A is the first input bit, B is the second input bit, Cin is the input carry bit, S is the output sum bit, and Cout is the output carry bit), where (I assume) the distribution over [A, B, Cin] is uniform, and where S and Cout follow deterministically from [A, B, Cin]. After training, the output of the circuit is computed from [A, B, Cin] by clamping [A, B, Cin] and sampling [S, Cout] given [A, B, Cin] using Gibbs sampling.

The alternative to this, which is examined in the paper, is to train individual XOR, AND, and OR gates in the same way and compose them into a complete binary adder circuit as prescribed by Section 3.

I think the paper has the potential to be a lot more transparent to the reader in explaining these concepts, which would avoid them spending quite a bit of time inferring meaning from figures.

I’m also confused by the presentation of the results. For instance, I don’t know what "log", "FA1", "FA2", etc. refer to in Figure 6. Also, Figure 6 is referenced in the text in the context of binary multiplication ("[...] is able to outperform a multiplier created just by training, as can be seen in Figure 6"), but presents results for addition and factorization only.

The way I see it, implementing reversible boolean logic circuits using RBMs is an artificial problem, and the key idea of the paper -- which I find interesting -- is that in some cases it appears to be possible to combine RBMs trained for sub-problems into larger RBMs without needing to fine-tune the model. I think there are interesting large-scale applications of this, such as building an autoregressive RBM for image generation by training a smaller RBM on a more restricted inpainting task. The connection to combinatorial optimization, however, is much less clear to me.

---

> ### Author Response · Authors · 2018-11-25
> **Reviewer Response**
>
> Thank you for your comments, we will be responding with specific comments to AnonReviewer4 here, and more general comments to the reviewer above.
>
> R4: “What’s less clear to me is what kinds of combinatorial optimization problems can be mapped onto the RBM *training* problem”
>
> The combination method we propose here can be applied to RBMs that are calculated by directly setting weights and by training individual sub units. We acknowledge there are pros and cons to both approaches; directly calculating the weights gives guarantees on probabilities and mixing rates, while training can produce a more compact, data, and computationally efficient model. Some algorithms will be more amenable to training, while others more amenable to directly calculating and setting weights, so we believe that this should be addressed on an algorithm by algorithm basis. We try to present one possible algorithm and a possible combination mechanism that we believe could work for others.
>
> R4: “The paper states that the problem of training "large modules" is "equivalent to solving the optimization problem", but does not explain how.”
>
> Training a full module to solve an optimization problem in the context presented here involves supplying samples from a large portion of the subspace that we are trying to model. Based on the results we have seen, we only achieve good performance once we have samples from >30% of the subspace (depending on the problem). In addition, the RBMs perform significantly better when trained on the full space we are trying to model.  We view this as the RBM creating an associative memory where it “memorizes” examples and recalls them afterward, and do not view this as a data and computationally efficient method of solving these problems.
>
> R4: An example is presented in Figure 3 but is not expanded upon in the main text. I’d like the authors to validate my understanding:
> An RBM is trained to implement a complete binary adder circuit by having it model the joint distribution of the adder’s inputs and outputs [A, B, Cin, S, Cout] (A is the first input bit, B is the second input bit, Cin is the input carry bit, S is the output sum bit, and Cout is the output carry bit), where (I assume) the distribution over [A, B, Cin] is uniform, and where S and Cout follow deterministically from [A, B, Cin]. After training, the output of the circuit is computed from [A, B, Cin] by clamping [A, B, Cin] and sampling [S, Cout] given [A, B, Cin] using Gibbs sampling.”
>
> Yes, your understanding is correct. We train on the joint density over inputs and outputs, and solving a problem amounts to clamping (conditioning) a subset of the units and sampling the remaining units via Gibbs Sampling. We have made an effort in the revision to make sure that this is more clear. In the case of solving factorization problem, we clamp some of the visible units to the integer we are trying to factor, and use gibbs sampling to get statistics for the remaining units conditioned on the output number.
>
> R4: I’m also confused by the presentation of the results. For instance, I don’t know what "log", "FA1", "FA2", etc. refer to in Figure 6. Also, Figure 6 is referenced in the text in the context of binary multiplication ("[...] is able to outperform a multiplier created just by training, as can be seen in Figure 6"), but presents results for addition and factorization only.
>
> We have presented results for addition and factorization in the main body of the paper, but refer to readers of the paper to the appendix where we have included a larger set of results. The results were omitted from the main body of the paper for the sake of brevity. The naming of units as “log” “FA1”, “FA2”, etc. are meant to represent the size of the base unit that was merged to create this larger unit, “log” referring to logical units (AND, XOR, etc.), “FA1” being 1 bit full adder, “FA2” being a 2 bit full adder, etc. we have made this clear in the figure caption.
>
> R4: The way I see it, implementing reversible boolean logic circuits using RBMs is an artificial problem, and the key idea of the paper -- which I find interesting -- is that in some cases it appears to be possible to combine RBMs trained for sub-problems into larger RBMs without needing to fine-tune the model.
>
> We also agree that there may be other applications to this type of merging of RBMs without further training, and we are working to look at those in greater detail. Invertible Boolean Logic provides a good test bed for this idea, and as explained above, we do believe it has a very intimate relationship with Boolean Satisfiability problems and other combinatorial optimization problems.

---

### Author Response · Authors · 2018-11-25
**General Comments to Reviewers**

We thank the reviewers for their detailed and thoughtful reviews! Based on their feedback, we uploaded a revised version of the paper.

We will be referring to the reviewers AnonReviewer2, AnonReviewer3, and AnonReviewer4 as R2, R3, and R4 respectively, and attempting to answer some of the comments that all of the reviewers had here.

R4: “I found the paper confusing at times. It is well-written from a syntactical and grammatical point of view, but some key concepts are stated without being explained, which gives the impression that the authors have a clear understanding of the material presented in the paper but communicate only part of the full picture to the reader.”

R3: “The paper is somewhat easy to follow and the figures are helpful. But the overall organization and flow of ideas can be improved significantly.”

R2: “The paper motivates the problem well, however, it is not well-written and at times it is hard to follow.”

All three reviewers mentioned that the paper did not read well, and could be confusing explaining concepts and not further detailing what is meant.  We have tried to be more clear in our methods, and have tried to restructure the paper slightly to ease understanding of the approach and concepts. In addition, we have went further into explaining certain important concepts (such as the exact method of combination) in greater detail.



R4: I have two issues with the chosen example: 1) the connection with combinatorial optimization is not clear to me, and 2) it’s not very well explained.

R3: “The term "combinatorial optimization" is used in a confusing way -- addition would not usually be called a combinatorial optimization problem.”

R2: “The term "Combinatorial optimization", which is used in the title and throughout the body of paper, sounds a bit confusing to the reviwer. This term is typically used in other contexts.”

We have also further in explaining our usage of the term “combinatorial optimization”. We view our combination method with invertible Boolean logic as a method of solving the Boolean satisfiability problem, which is the classic example of an NP-Hard combinatorial optimization problem, and many combinatorial optimization problems can be reduced to it (as shown in the paper by Karp et al. [3]) . We have shown arithmetic and integer factorization as a further application of the invertible Boolean logic that inherently solves the boolean satisfiability problem in its construction. This is further supported by showing the method of creating a full adder circuit through combinations of logic circuits.

We have revised the paper to further explain this concept, and refer the reviewers to the Section 3 “Approach”.

All three reviewers also commented on our choice of invertible Boolean logic to validate this approach.  We note that the integer factorization problem is in NP, and the Boolean satisfiability problem (which is solved within our invertible Boolean logic formulation) is also an NP-Complete problem. The reviewers also suggested using this approach on a more complex task such as TSP. We argue that integer factorization, invertible Boolean logic, and Boolean satisfiability can be harder in some ways than TSP, as integer factorization is an “all or nothing” problem, where the solution is either completely correct or completely incorrect.  Our choice of invertible Boolean logic was due to the simple and intuitive factorization of the problem into smaller sub problems. We also note (as mentioned above, and the revision of the paper) that NP-Complete formulations of the TSP problem can be reduced to a Boolean satisfiability problem ([2, 3]). We have also suggested how we could directly implement a TSP problem in the RBM.

[1] Pierre. Bremaud. Markov Chains: Gibbs Fields, Monte Carlo Simulation, and Queues, volume 1.Springer New York, 1999. ISBN 9781441931313.

[2] Karp, Richard M. "Reducibility among combinatorial problems." Complexity of computer computations. Springer, Boston, MA, 1972. 85-103.

[3]Stephen A. Cook. The complexity of theorem-proving procedures. In Proceedings of the Third Annual ACM Symposium on Theory of Computing, STOC ’71, pp. 151–158, New York, NY, USA, 1971. ACM. doi: 10.1145/800157.805047.

[4] Jansen, Boris, and Kenji Nakayama. "Neural networks following a binary approach applied to the integer prime-factorization problem." Neural Networks, 2005. IJCNN'05. Proceedings. 2005 IEEE International Joint Conference on. Vol. 4. IEEE, 2005.

---

### Meta-Review · Area_Chair1 · 2018-12-13
**Lack of clarity and justification for the final task**

**Confidence:** 5
**Recommendation:** Reject

**Metareview:**

Dear authors,

Thank you for submitting your work to ICLR. The original goal of using smaller models to train a bigger one is definitely interesting and has been the topic of a lot of works.

However, the reviewers had two major complaints: the first one is about the clarity of the paper and the second one is about the significance of the tasks on which the algorith is tested. For the latter point, your rebuttal uses arguments which are little known in the ML community and so should be expanded in a future submission.